# Study on Characteristics of Particulate Matter Resuspension in School Classroom through Experiments Using a Simulation Chamber: Influence of Humidity

**DOI:** 10.3390/ijerph18062856

**Published:** 2021-03-11

**Authors:** Sunghee Cho, Gahye Lee, Duckshin Park, Minjeong Kim

**Affiliations:** 1Transportation Environmental Research Team, Korea Railroad Research Institute, Uiwang 16105, Korea; hee719@krri.re.kr (S.C.); gahya1129@krri.re.kr (G.L.); dspark@krri.re.kr (D.P.); 2Department of Environmental Science and Engineering, Kyung Hee University, Yongin 17104, Korea

**Keywords:** particulate matter, resuspension, relative humidity, resuspension factor, occupants’ activity, classroom

## Abstract

Resuspension of particulate matter (PM) in classrooms, which increases the risk of negative impact on student health from exposure to PM, is influenced by humidity level in the indoor environment. The goal of this study is to investigate the properties of PM resuspension in accordance with relative humidity through classroom test chamber experiments. In actual classrooms, it is challenging to control factors influencing resuspension. Therefore, the classroom chamber that reflects the environment of elementary school classroom (e.g., structure, floor material) is used in this study. The humidity of the classroom chamber is adjusted to 35%, 55%, 75%, and 85% by placing it inside a real-size environmental chamber, which allows artificial control of climatic conditions. At the respective humidity conditions, PM resuspension concentration and resuspension factor caused by occupant walking across the classroom chamber are analyzed. The results show that both of the resuspension concentration and resuspension factor reveal a linear negative correlation to humidity increase. Furthermore, coefficient of determination (R^2^) indicating goodness-of-fit of the linear regression model between the resuspension concentration and humidity is 0.88 for PM_10_ and 0.93 for PM_2.5_. It implies that accuracy of the regression model for estimating PM_10_ and PM_2.5_ resuspension concentrations is 88% and 93%, respectively.

## 1. Introduction

Because people of our day stay indoors a long time, management of indoor air quality is closely related to occupants’ health. Teenage students in particular are reported to spend most of their time indoors [1]. As school classrooms are where these students spend the second-longest time in a day, the air quality within these spaces is considered an important source of health risks to students [2,3]. In addition, a decrease in indoor air quality is found to be associated with an increase in respiratory or cardiovascular disease. Teenagers do not have fully developed immune systems while they also have large respiratory volumes per kilogram of body weight. For this reason, they are more vulnerable to the influence of indoor air pollution than other age groups [4,5,6]. In this regard, the indoor air quality of school classrooms is emerging as an important factor in student health.

While there are a variety of causes of particulate matter (PM) emissions in classrooms, such as student activity (including walking and running), cleanliness of the floor, or the use of chalk for blackboards, PM resuspension from the floor is a major source [7,8,9,10,11]. In most cases, resuspended PM is inhalable and 1–10 µm in diameter. Shorter students might inhale the resuspended PM, or PM might settle on the surfaces of textbooks or stationery, increasing the risk of exposure [9,10]. In light of this, we need to identify the characteristics of PM resuspension to protect the health of students in classrooms.

The intensity of occupant activity, characteristics of the flooring (such as floor material and roughness of floor surface), and humidity are known as major factors influencing PM resuspension [12,13,14]. Of particular note is that high humidity causes capillary coagulation between particles and increases adhesion of their surfaces, hindering resuspension [15,16]. Accordingly, there have been many recent studies on PM resuspension under various humidity conditions [9,10,17]. Rosati et al. [9] measured PM resuspension from carpets of test house facilities at humidity levels of 20%, 40%, and 80%. The authors noted the resuspension factor decreased as humidity increased and larger particles were more influenced by humidity. Zheng et al. [10] measured PM resuspension from carpets by changing experimental room humidity from 50% to 80%. Kim et al. [17] measured PM resuspension from glass surfaces by changing wind tunnel humidity from 7% to 78%. The authors identified that resuspension began decreasing at humidity levels between 50% and 60%. However, these previous studies have limitations in that they did not reflect aspects of actual classroom environments such as floor material and structure (such as doors and windows). In this regard, we need a PM resuspension study in an environment reflecting actual classroom characteristics. However, if we conduct a study in an actual classroom, it will be challenging to control variables influencing resuspension, such as humidity, students’ activity, and dust mass loading of the floor [18]. Therefore, simulation chamber experiments are necessary in order to reflect the actual environments of classroom (e.g., structure, floor material) and to control the influential variables (e.g., dust mass loading, ventilation rate). In this way, the present study has a novelty that a classroom simulation chamber which reflects the actual environment, such as its structure and floor material, is produced and used for PM resuspension experiments. The major objectives of this study are (1) the measurement of PM resuspension under various humidity conditions through the classroom chamber experiments and (2) the analysis of resuspension factor, defined as the ratio of airborne PM mass above resuspension site to its surface mass, depending on humidity conditions.

Temperature and humidity of indoor spaces such as a classroom are found to be influenced by outdoor air [19,20]. Nguyen et al. [20] found that a building’s indoor and outdoor humidity had a correlation coefficient of 0.6, which means a moderate positive correlation. Based on these findings, the classroom chamber’s outdoor humidity was changed in this study using a real-scale environmental chamber that can artificially realize weather conditions such as temperature, humidity, and sunlight. The classroom chamber was placed inside the real-scale environmental chamber. The environmental chamber’s humidity was then adjusted, which means a change of outdoor humidity for the classroom chamber. Then, when indoor humidity of the classroom chamber changed following changes in the outdoor humidity, PM resuspension was measured. In this way, this study has a novelty that the PM resuspension was measured by realizing a classroom’s indoor humidity changes as influenced by outdoor humidity by using the real-scale environmental chamber.

Outlines of this paper are as follows. In Section 2, the previous research on PM resuspension under various conditions are introduced, and then, motivation of the present study is introduced. In Section 3, the specifications of classroom chamber and real-scale environmental chamber are explained. Then, the proposed methods for measuring PM resuspension under various humidity conditions are introduced. Section 4 presents the results for PM resuspension obtained from the classroom chamber. Finally, the conclusions of this article are addressed.

## 2. Literature Review on PM Resuspension Experiments

Occupant activity, flooring properties, surface dust loading, and humidity are known as major factors influencing PM resuspension. Recently, several studied on the PM resuspension experiments under various influential factor conditions were reported [9,10,17,21,22,23,24].

Rosati et al. [9] measured PM resuspension from carpets of test house facilities at humidity levels of 20%, 40%, and 80%. The authors noted the resuspension factor decreased as humidity increased and larger particles were more influenced by humidity. Serfozo et al. [22] evaluated PM resuspension rate depending on surface dust loading and walking properties (e.g., pathway and speed). The resuspension rate was independent on the walking properties, on the other hand, it increased at higher surface dust loading. Tian et al. [23] characterized walking-induced PM resuspension as a function of flooring type, relative humidity and surface dust loading using a consistent resuspension mechanism. They identified that the humidity revealed a conflicting influence on PM resuspension depending on the flooring type: on carpet the PM resuspension enhanced with the humidity increase, whereas on hard flooring the opposite effect was observed. Kim et al. [17] analyzed PM resuspension from glass surfaces by changing wind tunnel humidity from 7% to 78%. The authors identified that resuspension began decreasing at humidity levels between 50% and 60%. Benabed et al. [24] measured PM resuspension using a full-scale wooden chamber 2.5 m (L) × 2.5 m (W) × 2.5 m (H) in size. The authors noted that, for all particle sizes, resuspension fraction for hardwood flooring is larger than that for linoleum flooring. However, the previous studies have limitation that the experimental facilities did not reflect the aspects of actual classroom environments, such as the installed location and number of doors and windows. Therefore, in the present study, a classroom test chamber which reflects the actual environments of classroom (e.g., structure, floor material) and controls the influential factors (e.g., occupant walking pace, dust loading) is used. Hence, the test chamber experiments on PM resuspension by taking the actual classroom environments into account is the motivation of this study.

## 3. Materials and Methods

A proposed procedure for measuring PM resuspension caused by the occupant activity under various humidity conditions is as follows. First, a classroom simulation chamber was constructed 9 m (L) × 3 m (W) × 3 m (H) in size, equivalent to one-third of an actual elementary school classroom. Then, this classroom chamber was placed inside the real-scale environmental chamber 34 m (L) × 6 m (W) × 6 m (H) in size. Based on South Korea’s weather data over the recent three years, the outdoor humidity conditions of the classroom chamber were set at 35%, 55%, 75%, and 85% or above. Next, an occupant whose body weight is 46 kg, similar to the average body weight of a sixth grader, walked across the chamber in one direction. The occupant walked at a pace of 2 m/s, similar to the average pace of an elementary school student in South Korea [25,26]. Finally, resuspension concentrations of each PM size were measured when the occupant walked across the chamber, and then the author quantitively compared the influences of humidity on PM resuspension by calculating resuspension factor. Details on each part of the proposed procedure are explained in the following subsections.

### 3.1. Classroom Simulation Chamber

A classroom simulation chamber that is one-third the size of an elementary school classroom in Suwon City, Korea was created. Table 1 and Figure 1 show the design specifications and layout of this classroom chamber. Hinged doors were set at the front and back on the side of the chamber to reflect the entrances of actual classrooms. The chamber’s windows were the same in number, size, and location as those in an actual classroom. In addition, a sliding door was installed at the front and rear of the chamber, respectively, to allow the occupant access during the experiment. Such sliding doors were required to reduce the flow rate and PM concentration changes caused by opening or closing a door. A classroom floor is usually made of wooden decor-tile, imitation-stone terrazzo tiles, or linoleum tiles, and wooden decor-tiles were used in this classroom chamber.

To measure PM resuspension concentrations from the classroom chamber, aerosol spectrometers (Grimm Aerosol Technik, Ainring, Germany) that use light scattering measurement were employed. This instrument divides particles that are 0.25–35 µm in diameter into 31 channels and then measures count concentrations and mass concentrations for each diameter every 6 s [27]. In consideration of the respiratory tract positions of elementary school students when they sit in chairs, instruments were set at both 50 cm and 70 cm above the floor. To measure indoor and outdoor temperature and humidity of the classroom chamber, a HygroFlex3 digital transmitter (Rotronic, Bassersdorf, Swiss) was used. Its sensor measures temperatures between −40 °C and 60 °C and humidity between 0% and 100% every 1 s [28]. The indoor and outdoor temperature and humidity of the classroom chamber was monitored through four transmitters set on the chamber wall and one transmitter on the exterior.

### 3.2. Real-Scale Environmental Chamber

To adjust the classroom chamber’s outdoor humidity, a real-scale environmental chamber was used to realize weather conditions artificially. This weather environment test equipment is certified by the Korea Laboratory Accreditation Scheme (KOLAS) under the Korean Agency for Technology Standards of the Ministry of Trade, Industry, and Energy. It can simulate weather conditions such as temperature, humidity, sunlight, and freezing. Table 2 and Figure 2 show the operating specifications and appearance of the real-scale environmental chamber. The environmental chamber’s temperature for PM resuspension experiments was set at 25 °C, and humidity changed from 35% to 55%, 75%, and 85% or above.

### 3.3. PM Resuspension Experiment Methods

This study aimed to measure PM resuspension by changing indoor humidity following changes in outdoor humidity. Figure 3 is a schematic diagram of this experiment. The classroom chamber’s outdoor humidity was changed by placing the chamber in the real-scale environmental chamber and then changing the environmental chamber’s humidity. The PM resuspension experiment was conducted in the following steps: (1) PM was dispersed in the classroom chamber and then allowed to settle, (2) outdoor and indoor humidity of the classroom chamber was set, (3) the occupant walked in the classroom chamber and then PM resuspension concentrations were measured, and (4) the resuspension factor was calculated.

#### 3.3.1. Dispersing PM in Classroom Chamber and Allowing It to Settle

A solid aerosol generator (TOPAS GmbH, Dresden, Germany) was used to disperse PM in the classroom chamber. The solid aerosol generator aerosolizes dry and easily flowing substances such as solid particles and powders at a designated feed rate [29]. A1 ultrafine test dust (Powder Technology INC., Arden Hills, MN, USA) was used as the PM for this experiment. This standard test dust is 2.5 µm in central diameter and is used in efficiency tests of filters and air purifiers. Table 3 shows the chemical composition and physical characteristics of the A1 ultrafine test dust. In the classroom chamber, the test dust was dispersed from 2 m above the floor, and an air blower and four fans were used to uniformly disperse it. The air blower and fans were turned off after the dust was dispersed, and the author allowed 24 h for the dust to settle on the floor of the classroom chamber.

The dust surface concentration settling on the floor is known as a factor influencing resuspension concentrations [12]. For this reason, the volume of PM settled on the chamber floor was calculated in the following way:

(1) Filters (Whatman, Buckinghamshire, UK) were placed in the classroom chamber at an even distance;

(2) Test dust was dispersed in the classroom chamber and allowed to settle;

(3) Filter weights were measured;

(4) Dust surface concentration of the classroom chamber was calculating using Equation (1)
(1)$$\mathrm{dust}\;\mathrm{surface}\;\mathrm{conc}.\;=\;\frac{m_{after}-m_{before}}{\pi r^{2}}\;\;\left[g/m^{2}\right]$$

In Equation (1), *m_after_* and *m_before_* refer to the filter weights before and after dispersing and settling of the test dust, and *r* refers to the filter radius.

In this study, the classroom chamber’s dust surface concentration was set at 0.3 g/m^2^, which is the average of previous studies [9,13]. Consistent PM conditions on the chamber floor were ensured during the resuspension experiments by checking the dust surface concentration at every experiment.

#### 3.3.2. Setting Indoor and Outdoor Humidity of the Classroom Chamber

Weather conditions in Seoul and South Korea’s Gyeonggi Province over the most recent three years were analyzed to set the classroom chamber’s outdoor humidity conditions (See Table 4). Irrespective of the season, the region’s average humidity was 55% on non-rainfall days and 75% on rainfall days. Moreover, humidity reached as high as 90% on heavy rainfall days such as during the rainy season. Based on these findings, outdoor humidity was set at 35% for days with a dry weather warning, 55% for non-rainfall days, 75% days for rainfall days, and 85% or above for days with high humidity. Each outdoor humidity condition was set by adjusting the steam control valve of the real-scale environmental chamber. The classroom chamber’s indoor humidity was affected by the outdoor humidity, and a PM resuspension experiment was performed after waiting for 3 h to allow the indoor humidity to stabilize. According to enforcement regulations of School Health Act in South Korea, a classroom’s indoor temperature is maintained above 18 °C and below 28 °C. In accordance with this regulation, the test temperature was set at 25 °C during the experiments. Please note that a ventilation (including mechanical and natural) was not applied during the experiment, that is, no effect of the ventilation on PM resuspension. It means that a ventilation rate which influences transport and distribution of the resuspended PM was equal to 0.

#### 3.3.3. Occupant Walking and Measurement of PM Resuspension Concentrations

After the classroom chamber’s humidity was set according to the designated conditions, the occupant walked across the chamber. PM resuspension concentrations were then measured. The step-by-step details were:

(1) The occupant entered the classroom chamber and then waited for 5 min. This process was required to reduce changes in indoor PM concentrations caused by opening and then closing of the entrance door.

(2) The occupant walked across the chamber once at a pace of 2 m/s.

(3) The occupant exited the chamber after 10 min.

(4) When PM concentration in the classroom chamber had stabilized, the above steps were repeated.

Changes in PM_10_ and PM_2.5_ concentrations were measured with aerosol spectrometers (Grimm Aerosol Technik, Ainring, Germany) installed in the classroom chamber. The measured PM resuspension concentrations and durations were then analyzed (See Figure 4).

Resuspension concentration (µg/m^3^): PM concentration increased after occupant had walked

Resuspension duration (minute): Time spent to reach the resuspension concentration

#### 3.3.4. Occupant Walking and Measurement of PM Resuspension Concentrations

Lastly, PM resuspension factor was calculated under each humidity condition in the classroom chamber. Resuspension factor is the ratio of the measured airborne PM mass above the resuspension site to its surface mass and calculated with Equation (2).
(2)$$E_{Dp}=\frac{C_{resuspension}\times V_{room}}{C_{loading}\times A_{walk}}$$

In Equation (2), *E_Dp_* refers to resuspension factor (in µg/µg), *C_resuspension_* the resuspension concentration (in µg/m^2^), *V_room_* the volume of the classroom chamber (in m^3^), *C_loading_* the dust surface concentration (in µg/m^2^), and *A_walk_* the area in the chamber traversed by the occupant (in m^2^) [9].

Influences of humidity on PM resuspension were quantified by comparing the resuspension factor of the classroom chamber under each humidity condition. The correlation between humidity and resuspension factor was analyzed using a regression line.

## 4. Results and Discussion

### 4.1. PM Resuspension Characteristics under Various Humidity Conditions

Table 5 shows the variations of humidity inside the classroom chamber after influence from the changes in outdoor humidity. It is important to note that outdoor humidity for the classroom chamber was equal to humidity in the real-scale environmental chamber, since the former was placed inside the latter, which in turn allowed for artificial control of humidity between 5% and 95%. Even though the indoor humidity was relatively lower than outdoor humidity, indoor humidity tended to increase as outdoor humidity did so. This coincides with a previous study [20], which showed that indoor humidity has a modest correlation to outdoor humidity. This result indicates that the chamber experiment in this present study well reproduces the variation of indoor humidity from the effect of the outdoor humidity.

Figure 5 shows changes in PM_10_ concentrations in the classroom chamber under each humidity condition. The experiment was repeated 5 to 7 times under each humidity condition. At humidity levels of 35%, 55%, and 75%, PM_10_ concentrations surged immediately after the occupant walked across the chamber and then tended to reach the maximum concentration after about 1 min. On the other hand, at a humidity of 85% or above, PM concentration did not change much after the occupant walked across the chamber. This is because high humidity causes capillary coagulation between particles and hinders resuspension by increasing particle surface adhesion [10]. Similar to findings in previous studies, PM_10_ resuspension concentrations decreased as humidity increased (22 µg/m^3^ at 35%, 18 µg/m^3^ at 55%, 13 µg/m^3^ at 75%, and 4.6 µg/m^3^ at 85%).

Figure 6 shows changes in PM_2.5_ concentrations in the classroom chamber under each humidity condition. As was the case for PM_10_ concentrations, PM_2.5_ concentrations surged immediately after the occupant walked across the chamber and tended to reach maximum concentration after about 1 min. However, resuspension concentrations of PM_2.5_ were lower than those of PM_10_ (See Table 6). We can assume that such a difference was because humidity influenced particle surface characteristics, that is, the particles collide with each other causing coagulation, and at this time, smaller particles tend to agglomerate into larger particles [10].

Figure 7 displays the PM_10_ and PM_2.5_ resuspension concentrations as a function of humidity. It is important to note that *x*-axis of Figure 7 is the classroom chamber’s indoor humidity when its outdoor humidity was set at 35%, 55%, 75%, and 85%, respectively (for details, refer to Table 5). Resuspension concentrations and humidity reveal a strong negative correlation, which means an increase in humidity causes a linear decrease of resuspension concentrations. Regression equations between the indoor humidity and PM resuspension concentrations are expressed as:(3)$${\mathrm{PM}}_{10}=-0.4391\times Rh+39.563$$
(4)$${\mathrm{PM}}_{2.5}=-0.0684\times Rh+6.3157$$
where PM_10_ and PM_2.5_ mean resuspension concentrations, and *Rh* refers to indoor humidity of classroom chamber. Coefficient of determination (R^2^), which shows the goodness-of-fit of a regression equation, was 0.88 in the case of PM_10_ and 0.93 in the case of PM_2.5_. It indicates that the accuracy of regression equations between humidity and resuspension concentrations of PM_10_ and PM_2.5_ was 88% and 93%, respectively.

### 4.2. Comparison of PM Resuspension Factors under Each Humidity Conditions

Table 7 and Figure 8 display changes of PM_10_ and PM_2.5_ resuspension factors under each humidity conditions. Please note that the PM resuspension was measured by realizing the indoor humidity changes as influenced by outdoor humidity. Therefore, Figure 8 shows the variations of resuspension factors depending on both of indoor and outdoor humidity of the classroom chamber. Analogous to the correlation between humidity and resuspension concentrations, the resuspension factors were in a strong negative correlation with humidity. It coincides with the findings in previous researches [9,10,17]. The reason is considered that, at high humidity, contribution of van der Waals force between particles becomes small, and at this time, capillary force between them increases. It causes the increase of particle surface adhesion. Then, it makes the particles more difficult to detach, and therefore, suppresses the PM resuspension [10,13,23,30].

Equations (5) and (6) below show regression equations between the indoor humidity and resuspension factors of PM_10_ and PM_2.5_.
(5)$$E_{PM10}=-0.0002\times Rh+0.0188$$
(6)$$E_{PM2.5}=-3E-05\times Rh+0.003$$

In the above equations, *E_PM_*_10_ and *E_PM_*_2.5_ mean resuspension factors of PM_10_ and PM_2.5_, and *Rh* refers to classroom chamber’s indoor humidity. Coefficient of determination (R^2^) was 0.91 in the case of PM_10_ and 0.95 in the case of PM_2.5_, which means that the accuracy of regression equations between humidity and resuspension factors of PM_10_ and PM_2.5_ was 91% and 95%, respectively. These regression equations will enable estimating PM resuspension factors under various humidity conditions.

The resuspension factor in this study is lower than that in a previous study [9] (the factor ranged from 0.0022 to 0.0106 in this study, while 0.09 to 0.1 in the previous study). The activity difference between occupants of the two studies might have contributed to such results. This study’s occupant walked 10–13 steps fewer than the occupant in the previous study, who walked 300 steps. Because these two studies adopted different experiment conditions such as the level of occupant activity and the particle loading mass, it is hard to quantitatively compare the resuspension factor of the two studies. However, through the further experiments by using the classroom simulation chamber where disturbances are controlled, resuspension factor can be calculated and compared under various environments (such as a floor material or air-conditioning). For this reason, it will be possible to utilize the findings of this study to identify major factors in PM resuspension in classrooms and quantify the contributions of each factor.

## 5. Conclusions

An experimental study was conducted in a classroom simulation chamber to identify the characteristics of PM resuspension in such a classroom under various humidity conditions. This study differentiates itself in that it measured PM resuspension in indoor humidity changes that were influenced by outdoor humidity changes. This was achieved by placing the classroom chamber in a real-scale environmental chamber that could artificially realize weather conditions, and PM resuspension was measured when the classroom chamber’s indoor humidity had changed under the influence of the outdoor humidity. Resuspension factor, a ratio of the airborne PM concentration measured above the resuspension site to its surface concentration, was measured under each humidity condition. The PM resuspension concentration and factor tended to show a linear decrease as humidity increased. In particular, the accuracy of regression equations between humidity and resuspension concentration was 83% for PM_10_ and 93% for PM_2.5_; and that between humidity and resuspension factor was 91% for PM_10_ and 95% for PM_2.5_. For further study, validation of the regression equations between humidity and resuspension properties (at unexperimented humidity conditions or in actual classroom) will be carried out for the practical application of this study. Then, these results can contribute to estimation of particle resuspension in classrooms according to humidity conditions.

## Figures and Tables

**Figure 1 ijerph-18-02856-f001:**
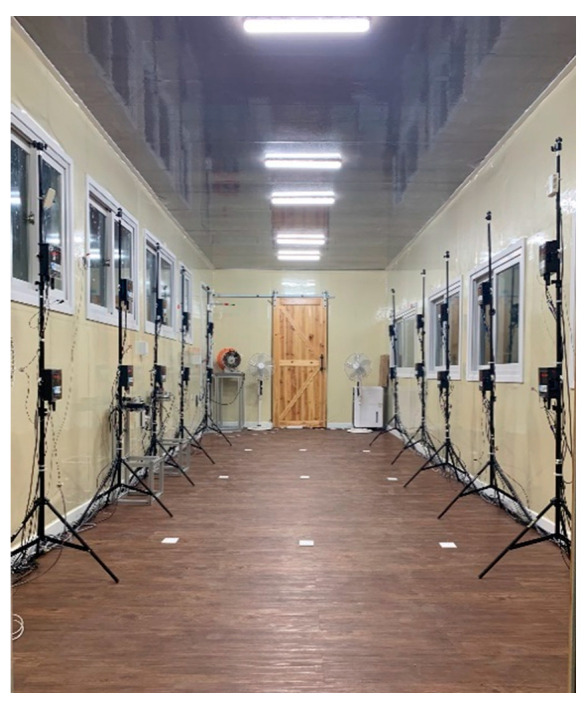
Inside the classroom chamber (Note: cube-shaped devices on the poles are simple particulate matter (PM) measurement devices but they were not used in this study).

**Figure 2 ijerph-18-02856-f002:**
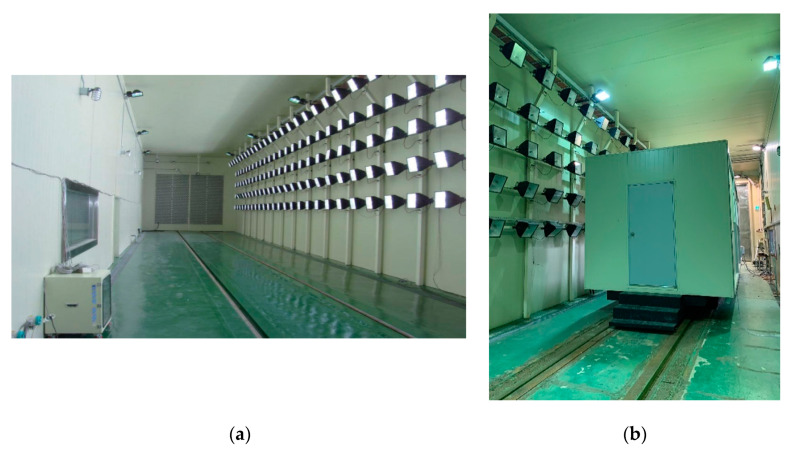
Real-scale environmental chamber: (**a**) inside the environmental chamber, (**b**) classroom chamber inside the environmental chamber.

**Figure 3 ijerph-18-02856-f003:**
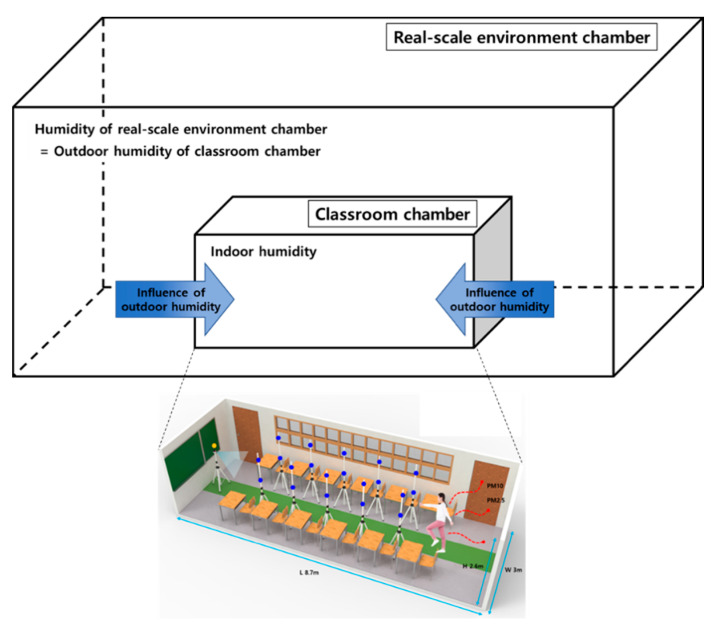
Schematic diagram of PM resuspension experiment in classroom chamber under various humidity conditions.

**Figure 4 ijerph-18-02856-f004:**
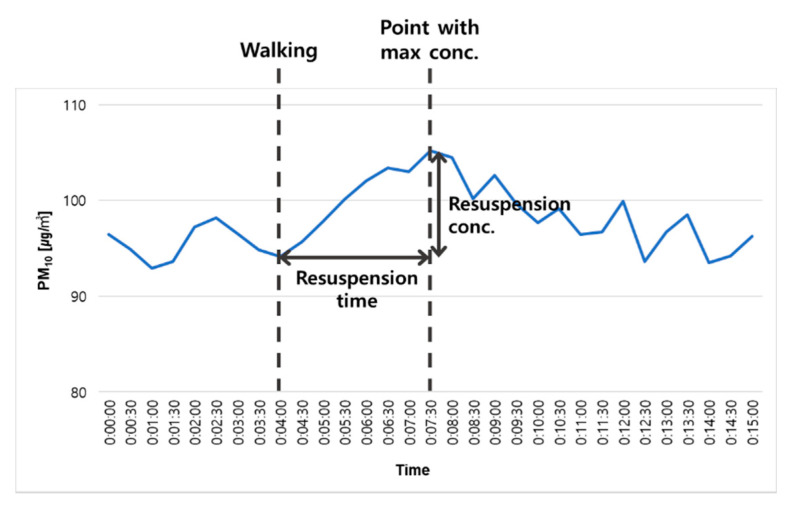
Example of analysis of PM resuspension concentration and time in the classroom chamber.

**Figure 5 ijerph-18-02856-f005:**
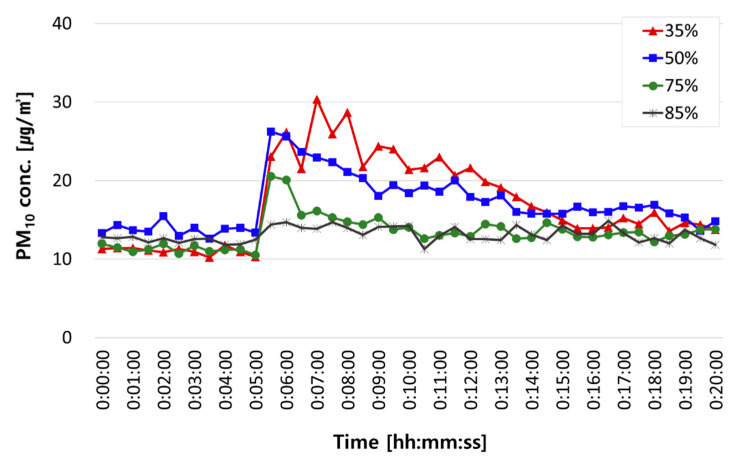
Changes in PM_10_ concentrations in the classroom chamber under each humidity condition: at 35%, 55%, 75%, and 85% humidity.

**Figure 6 ijerph-18-02856-f006:**
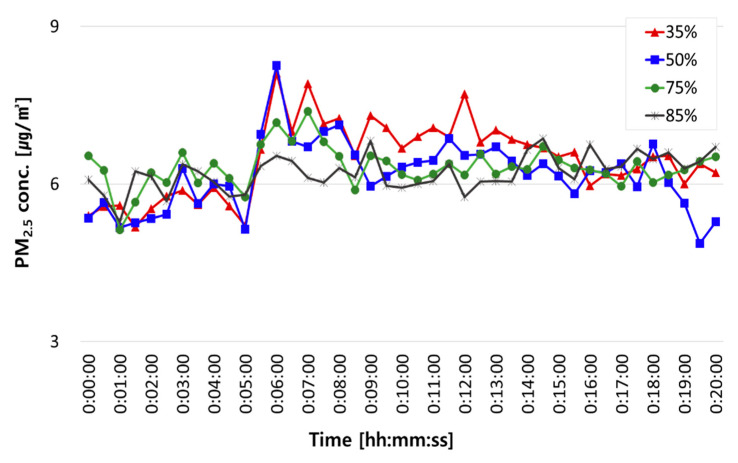
Changes in PM_2.5_ concentrations in the classroom chamber under each humidity condition: at 35%, 55%, 75%, and 85% humidity.

**Figure 7 ijerph-18-02856-f007:**
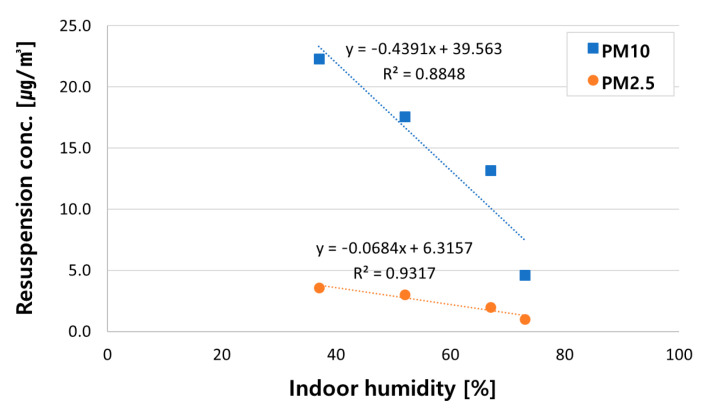
Comparison of PM resuspension concentration under each humidity condition.

**Figure 8 ijerph-18-02856-f008:**
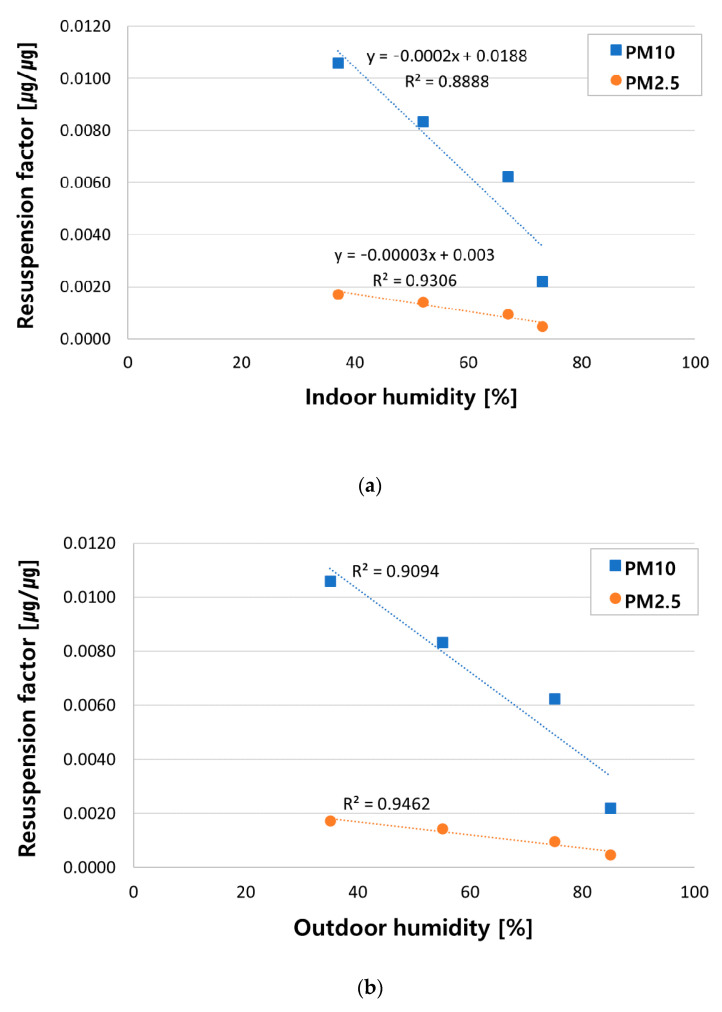
Comparison of PM resuspension factor under each condition of (**a**) indoor and (**b**) outdoor humidity of the classroom chamber.

**Table 1 ijerph-18-02856-t001:** Design specifications of the classroom chamber.

Item		Specifications	Characteristics
Size	Length	9 m	Same as actual classroom
	Width	3 m	One-third of actual classroom
	Height	3 m	Same as actual classroom
Entrance		900 mm × 2100 mm	Hinged door; installed at the front and rear on the sides of the chamber
Window to the corridor		1500 mm × 800 mm (2ea)	
Window to the schoolyard		1500 mm × 800 mm (4ea)	
Occupant’s entrance		900 mm × 2100 mm	Sliding door; installed at the front and rear of the chamber
Floor material		wooden décor-tile	

**Table 2 ijerph-18-02856-t002:** Operating specifications of real-scale environmental chamber.

Item		Specification
Chamber size		data
Humidity	Controllable range	5~95%
	Controllability	±5%
Temperature	Controllable range	−43~63 °C
	Controllability	<±1 °C
Freezing	Chamber temperature	<−5 °C
	Ice depth *	10 mm > at −10 °C
Solar irradiation		500~1200 W/m^2^
Wind speed		0~15 km/h

(note) * depth of ice which can be frozen on the floor and wall of the chamber.

**Table 3 ijerph-18-02856-t003:** Specification of the A1 ultrafine test dust (Powder Technology INC., Arden Hills, MN USA).

**(a) Chemical components**						
**Component**	**Quantity [%]**		**Component**		**Quantity [%]**	
SiO_2_	69–77		Al_2_O_3_		8–14	
Fe_2_O_3_	4–7		CaO		2.5–5.5	
K_2_O	2–5		Na_2_O		1–4	
MgO	1–2		TiO_2_		0–1	
**(b) Size distribution**						
**Size [µm]**	0.97	1.38	2.75	5.50	11.0	22.0
**Quantity [%]**	3–5	7–10	23–27	65–69	95–97	100

**Table 4 ijerph-18-02856-t004:** Weather conditions in Seoul and Gyeonggi Province over the most recent three years.

		Spring	Summer	Autumn	Winter
Non-rainfall days	Temperature (°C)	13.0	26.0	14.5	−1.4
	Humidity (%)	53.4	64.6	59.1	50.6
Rainfall days	Temperature (°C)	11.7	24.7	15.3	0.0
	Humidity (%)	69.3	77.4	72.2	63.1

**Table 5 ijerph-18-02856-t005:** Indoor and outdoor humidity for the classroom simulation chamber (inside the real-scale environmental chamber).

Indoor Humidity (%)	Outdoor Humidity (%)
37	35
52	55
67	76
73	85

**Table 6 ijerph-18-02856-t006:** PM resuspension concentrations in the classroom chamber under each humidity condition (the values in bracket represent the standard deviations).

[µg/m^3^ ]	35%	55%	75%	85% or Above
PM_10_	22.3 (±8.3)	17.6 (±6.8)	13.2 (±2.0)	4.6 (±1.6)
PM_2.5_	3.6 (±1.6)	3.0 (±2.0)	2.0 (±0.6)	1.0 (±0.2)

**Table 7 ijerph-18-02856-t007:** PM resuspension factors of the classroom chamber under each humidity condition.

[µg/µg]	35%	55%	75%	85% or Above
PM_10_	0.0106 (±0.004)	0.0033 (±0.003)	0.0062 (±0.001)	0.0022 (±0.001)
PM_2.5_	0.0017 (±0.0008)	0.0014 (±0.001)	0.0010 (±0.0003)	0.0005 (±0.0001)

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
