# Peer review of "Study on Characteristics of Particulate Matter Resuspension in School Classroom through Experiments Using a Simulation Chamber: Influence of Humidity"

_ijerph, 2021, doi:10.3390/ijerph18062856_

Round 1

Reviewer 1 Report

General comments

The manuscript entitled “Study on Characteristics of Particulate Matter Resuspension in School Classroom through Experiments using a Simulation Chamber: Influence of Humidity” by Cho et al. is mainly devoted to study the effect of the internal relative humidity variation on the particulate matter resuspension and concentration (both surface and volumetric) in a chamber representative of a school classroom. The Authors have analyzed this effect also in relation to the humidity of an external chamber (in which the school-classroom-chamber is included) considered representative of the external relative humidity. I believe that the study is really well designed and structured, then the discussed results are potentially very important and they can be generalized even for other indoor environments. Therefore, I recommend the publication of this manuscript in IJERPH after some minor revisions. My comments and suggestions consist mainly in some clarifications and in some improvements of the figures. Finally, I understand that the Authors are not English native speakers, but I suggest a linguistic revision of the manuscript.

Specific comments

Lines 81-92. Maybe this part could be moved to the next section because it is very technical.

Line 136. The meaning of “ice depth” in Table 2 is not so clear. Please explain.

Line 148. The sentence “What follows Figure 3 is a description of each step” does not correspond to what is reported in Figure 3. Please delete or rephrase.

Lines 153-175. The Authors should provide more details on the chemical composition and the physical characteristics of the aerosol particles dispersed in the classroom chamber.

Line 169. In my opinion, “dust loading” is not the correct word for the calculated parameter. “Dust surface concentration” is surely more correct in this case.

Lines 188-189. The Authors should also provide a short explanation for the selected temperature. I believe that a temperature of 25°C could be representative only of summer season.

Line 210. Replace “concentration” with “duration”.

Lines 218-220. This sentence is not properly correct because also Vroom and Awalk are included in the defined parameter.

Line 256. I suggest the Authors modifying the Figure 5 by replacing the four plots with a single plot showing the four temporal evolutions at different relative humidity by four different colors. In this way it is easier to see the main differences. In addition, I suggest modifying the PM10 concentration scale up to 40 or 45 µg/m3 so it is easier to see the temporal evolution differences between the different humidity conditions.

Lines 263-264. I suggest the Authors removing “converting a weak van der Waals force between the particles to a strong liquid bridge force”.

Lines 268-269. Why were the cases at 75% and 85% humidity not reported in Figure 6? My suggestion for Figure 5 to use one single plot (with different colors for different humidity conditions) is also valid for Figure 6. In addition, I suggest modifying the PM2.5 concentration scale up to 13 µg/m3 so it is easier to see the temporal evolution differences between the different humidity conditions.

Lines 273-277. I suggest the Authors also providing the regression equation and the other statistical parameters for the PM10 and PM2.5 concentration as a function of humidity because even these correlations could be potentially important, not only the PM resuspension factor.

Line 287. I suggest modifying Figure 7 by replacing histograms with scatterplots. In addition, it could be also interesting to repeat this figure with external humidity values, so you can evaluate the possible differences.

Lines 288-299. I suggest the Authors inserting some comments in the end of this section, mainly to explain the reason of the negative correlation between PM concentration/resuspension factor and humidity. In the current form of this section, this important explanation is completely missing.

Reviewer 2 Report

In this paper, authors  investigate the properties of particle resuspension in the classroom in accordance with relative humidity through test chamber experiments. The classroom simulation chamber used in the study was 1/3 the size of an actual classroom and placed inside a real-size environmental chamber which allows artificial control of climatic conditions (e.g., temperature, humidity). The humidity of the classroom chamber was adjusted to 35%, 55%, 75% and 85% accordingly.

The experimental design is accurate and well described. Results are clearly presented and conclusions are in aacordance with the presented results.

Reported references are wide and uptodate.

I suggest to publish this paper

Reviewer 3 Report

The first impact caused by the research is the minimalist content of the written paper. Given the quality of the experimental research, it was expected to have a more detailed analysis of the subject. The study is focused on the analysis of the properties of particle resuspension in the classroom under relative humidity through test chamber experiments. The Authors must try to enter deeper into the study so that the fundamentals of this investigation can be better understood. Otherwise, no matter how good the research is, the final objective of the study will end up being frustrated. Nevertheless, this is a very important issue concerning housing sciences and indoor air quality, and it is very related to the mission and aims of the Journal MDPI- IJERPH.

The abstract must be improved by adding the objectives of the study. Despite the study correlates the particle resuspension in the classroom with relative humidity through test chamber experiments, there is no evidence in the abstract of the main goals of the study. Is it related to a general study related to indoor air quality improvement in the classroom? Is it part of a deeper study that analyses other relevant parameters besides indoor airborne PM concentration? This research must be better framed so that the main objectives of this study can sound clear.

The Introduction Section (Section 1) is very little detailed. It is missing the structure of the article, mainly the sequence that the author chose to develop the study, the name of each Section and a brief description of its content, and the novelty of this research. The authors refer that the study measured PM resuspension by varying classroom indoor humidity in a classroom simulation chamber. However, what is the main objectives of this study? What is the final goal for this research? The topic of investigation is indeed very important regarding indoor air quality nevertheless it is not evident how do authors use these results for. How can this investigation be correlated with classroom environment improvement? The authors much work much more on that.

The authors didn't include in the research a specific section for “Literature review”.  The related works are presented in Section 1 (Introduction). The number of references (25) is short and is developed without relating the quoted studies to each other. Some new references should be added in a specific Section called “Literature review”, “Related works” or “State of the Art”. All new references should be related to the analyzed subject and it would be appreciated to quote some articles published in the International Journal of Environmental Research and Public Health regarding the investigation subject. There´s a lack of literature review for other countries' experiences concerning this analysis. It would be interesting to have this specific analysis for other places different from South Korea.

Section 2 regarding Materials and Methods includes subsections related to the “classroom simulation chamber”, “Real-scale Environmental Chamber”, and “PM Resuspension Experiment Methods”. The experimental study is very well detailed, however, is not clear what is the influence of outdoor air temperature on the results. What similar studies worldwide were already developed? What other variables must be controlled, namely the ventilation rate?

Section 3 refers to the discussion of the results. This study presents many strengths since it is a very analytical experimental research, however, there is some missing information that makes this investigation incomplete. This is particularly evident in this section. No results discussion regarding the influence of outdoor temperature and relative humidity on indoor hygrothermal conditions is presented, and Figures 5 to 7 must be better explained since they simply related relative humidity with PM resuspension. No extra analysis is done, and no correlation of this investigation with classroom indoor air quality is set. Some extra discussion on the results must be worked out.

Authors are invited to work much more on results discussion, adding some new blocks of text and some extra graphs and charts are preferable to organize information and better explain obtained correlations.

Section 4 is the conclusion section. Indeed, it is very poor. Conclusions must synthesize thoroughly all the outcomes of the investigation. Conclusions are not mere findings. I repeat, some stronger and meaningful conclusions are needed. The authors say that the investigation results can contribute to the estimate of particle resuspension in classrooms according to humidity conditions and to analyze the influence of particle resuspension on student health and safety during the long periods they are in the classroom. Nevertheless, nothing is said on how this analysis can contribute to student’s health and safety. I repeat, this is a robust experimental study that was not properly worked and correlated. The authors must do it more profoundly.

Round 2

Reviewer 3 Report

Reviewer comment 1:

The abstract must be improved by adding the objectives of the study. Despite the study correlates the particle resuspension in the classroom with relative humidity through test chamber experiments, there is no evidence in the abstract of the main goals of the study. Is it related to a general study related to indoor air quality improvement in the classroom? Is it part of a deeper study that analyses other relevant parameters besides indoor airborne PM concentration? This research must be better framed so that the main objectives of this study can sound clear.

Authors action:

The authors improved the abstract as advised. The abstract has now some quantification for the results of the investigation and refers to the main achievements of this research.

Reviewer comment 2:

The Introduction Section (Section 1) is very little detailed. It is missing the structure of the article, mainly the sequence that the author chose to develop the study, the name of each Section and a brief description of its content, and the novelty of this research. The authors refer that the study measured PM resuspension by varying classroom indoor humidity in a classroom simulation chamber. However, what are the main objectives of this study? What is the final goal for this research? The topic of investigation is indeed very important regarding indoor air quality nevertheless it is not evident how do authors use these results for. How can this investigation be correlated with classroom environment improvement? The authors much work much more on that.

Authors action:

Despite some improvement have been done to the introduction section, namely by taking out the related work and previous works in the subject to a separate Section, and by including the aim of this investigation, some missing questions must be answered namely regarding the novelty of this research.

Reviewer comment 3:

The authors didn't include in the research a specific section for “Literature review”.  The related works are presented in Section 1 (Introduction). The number of references (25) is short and is developed without relating the quoted studies to each other. Some new references should be added in a specific Section called “Literature review”, “Related works” or “State of the Art”. All new references should be related to the analyzed subject and it would be appreciated to quote some articles published in the International Journal of Environmental Research and Public Health regarding the investigation subject. There´s a lack of literature review for other countries' experiences concerning this analysis. It would be interesting to have this specific analysis for other places different from South Korea.

Authors action:

A new section regarding the literature review on PM resuspension experiments was included. Some new references were added.

Reviewer comment 4:

Section 2 regarding Materials and Methods includes subsections related to the “classroom simulation chamber”, “Real-scale Environmental Chamber”, and “PM Resuspension Experiment Methods”. The experimental study is very well detailed, however, is not clear what is the influence of outdoor air temperature on the results. What similar studies worldwide were already developed? What other variables must be controlled, namely the ventilation rate?

Authors action:

Despite some improvements that have been done, there is still a question to be answered: what other variables must be controlled, namely the ventilation rate?

Reviewer comment 5:

Section 3 refers to the discussion of the results. This study presents many strengths since it is a very analytical experimental research, however, there is some missing information that makes this investigation incomplete. This is particularly evident in this section. No results discussion regarding the influence of outdoor temperature and relative humidity on indoor hygrothermal conditions is presented, and Figures 5 to 7 must be better explained since they simply related relative humidity with PM resuspension. No extra analysis is done, and no correlation of this investigation with classroom indoor air quality is set. Some extra discussion on the results must be worked out.

Authors action:

Figures 5 to 7 are now better explained and some extra analysis concerning classroom indoor air quality is set. Some new results regarding the influence of outdoor temperature and relative humidity on indoor hygrothermal conditions are now presented.

Reviewer comment 6:

Authors are invited to work much more on results discussion, adding some new blocks of text and some extra graphs and charts are preferable to organize information and better explain obtained correlations.

Authors action:

New section 4 is now better detailed and reasoned.

Reviewer comment 7:

Section 4 is the conclusion section. Indeed, it is very poor. Conclusions must synthesize thoroughly all the outcomes of the investigation. Conclusions are not mere findings. I repeat, some stronger and meaningful conclusions are needed. The authors say that the investigation results can contribute to the estimate of particle resuspension in classrooms according to humidity conditions and to analyze the influence of particle resuspension on student health and safety during the long periods they are in the classroom. Nevertheless, nothing is said on how this analysis can contribute to student’s health and safety. I repeat, this is a robust experimental study that was not properly worked and correlated. The authors must do it more profoundly.

Authors action:

New Section 5 has been redrafted and outlines now some practical and meaningful conclusions, by highlighting the importance of the results. However, nothing is said about the way how the estimation of particle resuspension in classrooms according to humidity conditions, can influences students’ health and safety during the long periods they are in the classroom.
